# Pushing the Limits of Surgical Resection in Colorectal Liver Metastasis: How Far Can We Go?

**DOI:** 10.3390/cancers15072113

**Published:** 2023-04-01

**Authors:** Francisco Calderon Novoa, Victoria Ardiles, Eduardo de Santibañes, Juan Pekolj, Jeremias Goransky, Oscar Mazza, Rodrigo Sánchez Claria, Martín de Santibañes

**Affiliations:** Department of Surgery, Division of HPB Surgery, Hospital Italiano de Buenos Aires, Buenos Aires C1199, Argentina; francisco.calderonnovoa@gmail.com (F.C.N.);

**Keywords:** colorectal metastases, liver resection, parenchyma sparring surgery, liver transplant, ALPPS

## Abstract

**Simple Summary:**

Colorectal cancer is the third most common cancer worldwide, and approximately half of all diagnosed patients will develop liver metastases. The only curative option for colorectal liver metastases is resection. Over the last decades, liver resection for colorectal metastases has been in constant evolution, and the limits of what is considered possible are continuously being pushed forward, benefiting many patients. This article sought to review the advances in the field of colorectal liver metastasis surgery and serves as a source of information for surgeons and physicians who wish to learn about all the possibilities in store for patients with colorectal liver metastases.

**Abstract:**

Colorectal cancer is the third most common cancer worldwide, and up to 50% of all patients diagnosed will develop metastatic disease. Management of colorectal liver metastases (CRLM) has been constantly improving, aided by newer and more effective chemotherapy agents and the use of multidisciplinary teams. However, the only curative treatment remains surgical resection of the CRLM. Although survival for surgically resected patients has shown modest improvement, this is mostly because of the fact that what is constantly evolving is the indication for resection. Surgeons are constantly pushing the limits of what is considered resectable or not, thus enhancing and enlarging the pool of patients who can be potentially benefited and even cured with aggressive surgical procedures. There are a variety of procedures that have been developed, which range from procedures to stimulate hepatic growth, such as portal vein embolization, two-staged hepatectomy, or the association of both, to technically challenging procedures such as simultaneous approaches for synchronous metastasis, ex-vivo or in-situ perfusion with total vascular exclusion, or even liver transplant. This article reviewed the major breakthroughs in liver surgery for CRLM, showing how much has changed and what has been achieved in the field of CRLM.

## 1. Introduction

Colorectal cancer (CRC) remains the third most common cancer worldwide [1]. Although screening for CRC is practiced ubiquitously, approximately 30 to 50% of the patients will eventually develop either synchronous or metachronous liver metastasis [2]. The best option for patients with colorectal liver metastasis (CRLM) has always been surgery. Survival in these patients has shown improvement because of multidisciplinary board management [3,4,5] and very promising results with systemic chemotherapy [6,7], with 5-year overall survival (OS) ranging from 20–45% [8]. The eligibility criteria for curative surgery have undergone continuous modifications from the earliest reports of successful resections in CLRM over 40 years ago [9,10,11]. Advances regarding different ways to calculate and stimulate the future remnant, technically complex surgeries such as parenchyma sparing surgery (PSS), intraoperative ultrasonography (IOUS), and even ex-situ hepatectomy and liver transplant continuously extend the limits in surgical resection of patients with CRLM. This article reviewed the major advances in CRLM surgical resection and their impact on daily practice.

## 2. Calculating the Future Liver Remnant and Making It Grow: From Volumetrics to Portal Vein Embolization

The liver’s regenerative capacity has been well documented in the early 20th century [12]. This knowledge has led to the development of different types of liver resections, both anatomic and non-anatomic. Major hepatic resections have long been a feared procedure. On one hand, the highly vascularized anatomy of the liver very frequently complicates the procedure with major bleeding, which may require blood transfusions. This is now managed with restrictive fluid administration intraoperatively, as well as the use of intermittent clamping via the Pringle maneuver [13]. On the other hand, the most feared complication of major resections has always been post-hepatectomy liver failure (PHLF). Numerous studies have defined thresholds regarding the minimal amount of viable liver needed to avoid PHLF. For healthy livers, minimal FLR must represent approximately 20% of its total. However, thresholds for cirrhotic livers and patients who received chemotherapy tend to be set at 40% and 30%, respectively [14,15,16]. 

There have been numerous formulas and software tools developed to assess the amount of FLR after surgery. Common formulas rely on CT measurements of the remaining liver to calculate the total liver volume and the use of body measurements such as the body mass index or body surface area. A commonly used formula is standardized future liver remnant (sFLR) [17]. However, there are numerous reports of PHLF in spite of adequate volumetries. This is something to be expected, considering many patients present with chronic liver disease and/or prolonged chemotherapy treatment. These “volume-function” discrepancies have to be addressed in order to assure safe resections. The use of functional studies was developed to better assess the FLR. Hepatobiliary scintigraphy (HBS) using iminodiacetic acid marked with Tc-99 [18,19] and single-photon emission computed tomography (SPECT) [20,21] are the main methods developed to assess FLR functionality. 

The first hurdle to jump in liver surgery is clear: making the FLR good enough for patients who needed major resections. There is already sufficient evidence that shows left liver hypertrophy in cases of right side portal thrombosis in patients with hilar [22]. With this information, the first deliberate portal vein embolizations (PVE) were performed in 1990 by Maakuchi et al. [23]. Further reports showed that PVE was a safe, feasible, and effective procedure. Azoulay et al. in 2000 [24] analyzed 30 patients with PVE. Of these initially non-resectable patients, 63% were “rescued” using this technique, and survival did not differ significantly from initially resectable patients. Evidence shows that PVE can generate an FLR hypertrophy rate of 40% with a technical success rate above 99% and a clinical success of 96% [25]. Figure 1 shows a successful embolization by CT volumetrics, and Figure 2 shows the surgical results of the same patient.

However, PVE has encountered some setbacks. Hemoperitoneum, pneumothorax, hemobilia, arteriovenous fistulas, biliary complications and even migration of the embolization material to the FLR can complicate the procedure. Technical measures such as ipsilateral access and balloon occlusion were devised to minimize this risk. On the other hand, several reports have alerted the scientific community to a potential issue: the augmented portal flow and growth factors flowing to the FLR could generate growth in previously unnoticed metastasis [26,27]. Although in these reports segment IV veins were not systematically embolized nor did patients undergo preoperative chemotherapy, there are newer reports that support these findings. These patients with tumor growth in the FLR cannot be considered adequate candidates for surgical resection, precluding them from receiving curative treatment. 

It is clear that PVE had strong indications in a select group of patients with CRLM. However, patients with extensive bilateral disease were not suitable for PVE and therefore were not considered for curative intent surgery. Soon enough, different alternatives were devised for these cases.

## 3. Two-Staged Hepatectomy (TSH), Associating Liver Partition with Portal Vein Ligation for Staged Hepatectomy (ALPPS), and Mini-ALPPS: A Way around Extensive Bilobar CRLM

The main issue with bilateral extensive disease lies yet again in the FLR. When predicting an extreme resection and poor FLR, PVE is usually warranted. However, the tumor burden usually precludes PVE, with a higher chance of tumor growth in the FLR and poor outcomes. TSH was described in the early 2000’s [28,29,30] as an alternative for this subgroup of patients. Classically, TSH consisted of addressing the major hepatectomy in the first stage and clearing the FLR of its tumor burden at the second stage. The feasibility of TSH was approximately 60–80% [30,31], and OS of TSH patients was comparable at the time to patients who were initially resectable with a single hepatectomy (60% at 3 years, 40% at 5 years). Recent meta-analyses show acceptable oncological results for these patients, and variable second-stage completion [32]. However, in spite of the technical and oncological feasibility of TSH, there was still an unresolved issue: the high rates of PHLF. There are several reports in which PHLF reached almost 15% [30,31], with considerable mortality rates. 

With this problem far from being solved, there have been different attempts to develop new methods to address this. In 2012, Schnitzbauer et al. [33] presented the first series of patients in which an association of liver partition and portal vein ligation was performed for a staged hepatectomy, a procedure which would be later named “ALPPS” [34]. ALPPS promotes an extremely quick and efficient FLR hypertrophy, and even challenged the classical paradigm of major hepatectomies, which requires preservation of at least two contiguous segments with adequate inflow and outflow [35,36,37]. The combination of the liver injury and portal flow diversion to the FLR proved a spectacular option for quick hypertrophy. As mentioned, ALPPS combines an in situ liver partition and PVL for its first stage, as well as a cleanup of the FLR. The ligated hemiliver’s biliary tract and arterial flow must be preserved, and leaving it in situ assists the FLR as an accessory liver. Following the first stage, intersage volumetry is performed to evaluate FLR growth. For cases in which volumetry may be unreliable, functional studies have been used. HIBA index [38] has been developed for interstage evaluation of the FLR by Serenari and the Italian Hospital of Buenos Aires Group. A cutoff value of 15% can accurately predict PHLF.

After acceptable FLR is reached and the patient is still in good condition, stage two is performed with excision of the deportalized lobe. The mechanisms by which ALPPS’ hypertrophy is superior are yet to be completely elucidated, though the answer may lie in the interruption of communicating vessels between the hemilivers which may serve as an alternative for portal flow towards the deportalized liver, combined with the increased release of inflammatory mediators due to the transection [39].

The main issue with early reports of ALPPS was the high complication rates. There were high rates of biliary fistulas, bilomas, and sepsis. Mortality rates for the first series were staggering, ranging from 12% to almost 30% [33,40,41,42]. In 2014, the International ALPPS registry [43] showed results for 202 patients, 70% of these being for CRLM. FLR interstage growth was 80% in a mean of 7 days. Major complications (D-C IIIb or higher) were 28% for the whole cohort and 9% mortality, with a 98% completion of both stages. Complication rates were significantly lower in patients with CRLM than other malignancies. In spite of the astonishing results in terms of hypertrophy, PHLF still remained the main cause of mortality in the ALPPS. Possible explanations may be a poor correlation between volume and function due to edema and immature hepatocytes [44]. This showed that timing is of the essence, and volumetry does not necessarily correlate with adequate function. A critical point regarding ALPPS’s safety and feasibility was that the presence of major complications in stage 1 was a prognostic factor associated with mortality following stage 2 [45,46]. Several variations have been implemented by ALPPS advocates around the world in order to mitigate the impact of the first stage. Briefly, tourniquet ALPPS [47] included a vicryl tourniquet over the future transection line in order to completely ligate communicating vessels between the FLR and the soon-to-be-removed right lobe. Microwave ablation ALPPS [48] consists of a laparoscopic ligation of the right portal vein, followed by microwave ablation (MWA) utilizing several antenna probes in the future transection line, in order to generate an “avascular groove” that would separate the FLR from the embolized liver. Radiofrequency-assisted ALPPS shares the same concept with MWA-ALPPS, i.e., applying radiofrequency in the future transection line in order to achieve tissue necrosis and stimulation of growth without physical damage, thus limiting the number of complications in stage one. Partial-split ALPPS [49] is very similar to conventional ALPPS, with the difference that liver transection is not complete, and the in situ deportalized liver is only transected in 50–80% of cases. This was found to greatly reduce the morbidity of stage I and allow a higher stage II completion. Perhaps, the variation that finally managed to shift the paradigms was Mini-ALPPS [50]. The combination of partial transection and PVE managed to decrease complications for stage 1, making stage 2 safer. These new advances showed a dramatic decrease in morbidity and mortality reported by the ALPPS registry in 2017 [51]. With a total of 437 patients in the analysis, mortality rates showed great improvement, from 17% in the early period to 3.8% in 2015, a value in range of the established benchmarks for liver surgery [52]. Perhaps the main change that allowed for better results in ALPPS was the selection of younger patients with CRLM, which was identified as the population most likely to benefit from ALPPS. Minimally invasive variants of ALPPS were also associated with significantly less complications than traditional ALPPS. In 2019, benchmark values for ALPPS were published by the ALPPS registry group [53], as well as a preoperative ALPPS risk score to evaluate possible candidates [54]. Figure 3 shows the first stage of a combined Mini-ALPPS with a colonic resection, and Figure 4 shows the second stage in that same patient.

The choice between a TSH and ALPPS is not an easy task. Although both procedures are feasible for patients with extensive bilateral CLRM, ALPPS has been historically considered a “high-risk procedure”, most likely because of its excessive use and indications in its early developmental stages. High-quality evidence regarding superiority of one procedure over the other is scarce, mainly retrospective matching analysis, and only one randomized controlled trial so far (LIGRO) [55], which showed better resection rates for ALPPS over TSH (92% vs. 57%). Complications and mortality were similar in both groups with no statistical difference. A recent meta analysis by Moris et al. [32] showed more complications in the ALPPS patients rather than TSH, but no difference in liver-related mortality or oncologic results. As mentioned earlier, completion of both stages was higher in the ALPPS group over TSH. It would seem that ALPPS is slowly gaining acceptance worldwide with the enhanced results thanks to the implementation of less invasive variations, less aggressive first stages, and refined patient selection criteria. 

## 4. How Much Margin Do We Actually Need? Vascular R1 (R1vasc), PSS and IOUS

Providing the patient with adequate margins is one of the main principles in oncological surgery. However, this principle clashes with a liver surgery principle: sparing the highest amount of parenchyma as possible. This has several explanations. Firstly, a conservative resection greatly reduces the risk of PHLF and other complications associated with major resections. Secondly, a conservative resection does not preclude repeat procedures, which are common in CRLM, and may be limited by a prior major resection. Given this great dichotomy seen in CRLM surgery, it is no surprise that several groups have attempted to push the limits not only regarding resection size, but also, on the contrary, attempting to set limits regarding minimally acceptable margins and their outcomes. 

By the end of the 20th century, several reports showed that a 10 mm margin was no longer mandatory, and satisfactory results could be achieved with <10 mm margins [56,57,58]. In 2008, de Haas et al. [59] showed comparable results in R0 and R1 patients with colorectal metastatic disease, regarding OS, progression-free survival and disease-free survival. Perhaps the most important finding was that there was no difference in surgical site recurrence as one might have thought. Possible explanations may be related to the use of ultrasonic dissectors, which are thought to eliminate approximately 1–2 mm of viable tissue, and advances in chemotherapy. Newer and more effective regimes can assist in decreasing tumor burden preoperatively and micrometastasis as well. 

With this new evidence supporting macroscopically complete resections, different groups sought different ways to preserve viable hepatic tissue. Almost twenty years ago, the first reports by the Humanitas group regarding the use of IOUS showed that expert use could significantly decrease the need for major hepatectomies and their implications [60,61]. By using IOUS, they were capable of identifying and preserving communicating vessels, and most importantly, they were able to better define the relationship with the tumors with main vascular structures. This is where the concept of “vascular R1” (R1Vasc) arose, and differentiates from “parenchyma R1’’. R1Vasc is considered when the tumor is dettachted from a vascular structure, either Glissonian or a hepatic vein. Subsequent studies from Torzilli et al. showed that R1vasc had comparable results to R0 in both colorectal metastasis and hepatocellular carcinoma [62,63]. As to be expected, parenchyma R1 had poorer survival compared with R0 in these cohorts. The strongest theory supporting these findings is that vascular structures limit tumoral spread. Figure 5 shows a patient with a PSS and a R1 Vasc, and Figure 6 shows a PSS surgery in which partial vascular resection and replacement was necessary for the right hepatic vein.

The use of IOUS and R1 Vasc has many potential benefits: To begin, it may limit the need for major hepatic resections by avoiding great vessel resection or identifying communicating vessels which might salvage the otherwise healthy parenchyma drained by a mayor hepatic vessel which must be sacrificed due to tumor involvement. Following this same principle, IOUS may allow for preservation of adjacent sectors if portal flow is adequate and good drainage can be obtained by identifying communicating vessels. Finally, IOUS and the R1 Vasc concept might allow patients who used to be deemed unresectable a chance for surgery, with acceptable results. 

There followed numerous reports favoring PSS over anatomical major hepatectomies from across the globe. These were coherent with initial reports, showing that PSS was better in terms of patient safety, but also found no difference in oncological short- and long-term results [64,65,66,67]. In 2016, Mise et al. [68] compared 300 patients who underwent either a PSS or a major hepatectomy for small (<30 mm) CRLM. There was no difference in RFS between groups, but there was significantly higher 5-year survival in patients who underwent PSS. There were also significantly more patients who received a repeat hepatectomy upon liver-only recurrence in the PSS vs. the major hepatectomy group, showing that not only does survival favor PSS, but also that PSS does not preclude a repeat hepatectomy, expanding the amount of patients that can benefit from repeat procedures. A recent metaanalysis compared over 7000 who received either a PSS or major hepatectomy throughout 18 different studies [69]. PSS was superior to major hepatectomies in terms of perioperative morbidity, and there was no difference in terms of OS, RFS, and surgical margins. This means that PSS can safely acquire R0 margins without oncological compromise of the patient while avoiding major hepatectomies.

PSS, IOUS, and R1 Vasc push the limits in terms of what we perceive as unresectable disease, and may benefit many patients. However, there are a few issues, including the technical difficulties and expertise necessary for the correct use of IOUS, the extended surgical times, the apparent “setback” regarding the approach (PSS most of the time requires open surgery), and the lack of randomized controlled trials comparing PSS to major hepatectomies.

## 5. Synchronous CRLM: Two Birds with One Stone?

Depending on the series, up to 30% of patients with CRC may present with synchronous metastasis at diagnosis. These patients tend to have an aggressive biology and worse outcomes than patients with metachronous metastasis [70,71]. Classical management dictates a staged approach: control of the primary tumor, followed by chemotherapy, and then hepatectomy for CRLM. Another option for a staged approach is the “liver-first” approach. However, one-third of patients with initially resectable disease are unable to complete the treatment [72] because of tumor progression or surgical complications. These patients present a challenge for the multidisciplinary team (MDT). Teams must balance out the need for a rapid procedure with the risks implied in the two procedures needed. Each approach has its theoretical benefits: a classical colorectal-first approach may prevent tumor complication with occlusion or perforation, which is a known independent factor for peritoneal carcinomatosis and poor survival, and may also avoid further tumoral spread by controlling the primary tumor. A liver-first approach has a theoretical advantage, considering that the presence of liver metastasis is a marker of poor outcomes in patients with CRC. The simultaneous approach attempts to salvage the third of patients who never complete both surgeries and miss their chance at curative treatment. A recent Swedish study compared surgical and oncological outcomes in a large retrospective series of patients with either a simultaneous, liver-first, or colorectal-first approach [73]. There were no significant differences in surgical complications (15% for the simultaneous approach vs. 9.8 and 10% for liver-first and colorectal-first approaches) or mortality, which ranged from 0.6 to 1.5% in the three groups. It should be noted that there was a clear preference for minor hepatectomies in the simultaneous group, most likely because of selection of better candidates, which may account for the lack of statistical difference in complications. Five-year OS for all groups was similar, with slightly better results for the simultaneous group (48.9% vs. 44.2% and 44.3%), with no statistical difference whatsoever. There have been other retrospective studies and meta-analyses regarding the topic [74,75,76,77,78,79], which highlight the heterogeneity in metastatic disease as a main bias in these cohorts, making them difficult to compare. In 2020, the French study group METASYNC [80] recruited a total of 85 patients and allocated them to either a simultaneous approach (39 patients) or a colorectal-first approach (46 patients) and compared the results. There was no difference in major complications between groups, and five-year OS and disease-free survival (DFS) also tended to favor the simultaneous approach (5.9 and 1.3 years for the simultaneous approach and 3.9 and 1.0 years for the colorectal-first approach) with statistical significance. Simultaneous surgery was determined to be an independent factor for better OS in this study. They concluded that the simultaneous approach is a valid option for initially resectable synchronous CLRM; however, there is still a need for firmer evidence regarding safety and oncological benefits compared with the staged approaches. Figure 2 shows the surgical specimens of a simultaneous procedure.

The simultaneous approach may be a good option for patients who present synchronous CLRM. However, because of heterogeneity in studies as well as different clinical scenarios (asymptomatic vs. symptomatic primary tumor for example), there is still controversy regarding the indication of the simultaneous approach. Currently, this approach is recommended in patients with oligometastatic disease in which minor hepatectomies are predicted [81]. The successful combination of two large surgeries such as a colectomy and a hepatectomy into one surgery is another barrier that has been broken in the field of CRLM.

## 6. Minimally Invasive Liver Surgery (MILS) and Its Place in CRLM Surgery

Minimally invasive surgery is well established to have benefits over open surgery, such as complication reduction, pain management, length of hospitalization, and a quicker return to common life. MILS has been established as the “gold standard” for many hepatic diseases, both benign and malignant [82]. However, MILS for CLRM has various potential setbacks, such as the technical complexity required for specific types of resection, the elevated risk of uncontrolled bleeding, the possibility of air embolism into the IVC [83], and the need for complex technology which requires extensive training, such as laparoscopic IOUS. The need for trained surgeons and high-volume centers explains why MILS has had a slow introduction in the field of liver surgery. In 2015, an expert panel defined MILS as an adequate alternative for CRLM surgery [84].

There are a myriad of retrospective articles in the past 20 years showing superior results of MILS over open hepatectomy in terms of blood loss, LOH, complication rate, and comparable OS and DFS rates [85,86,87,88,89,90]. However, these studies were retrospective in nature, and many presented selection bias favoring MILS with smaller and usually more accessible tumors. In recent years, two RCTs have compared open surgery vs. MILS. In 2018, OSLO-COMET RCT compared MILS vs. open surgery, seeking to determine superiority of MILS [91]. Patients with CRLM amenable for PSS were included, except those who were programmed to have simultaneous surgery. Of 280 total patients, 133 were assigned to the MILS group and 147 to the open surgery group. Patients in the MILS groups had significantly less clinically relevant complications (19% vs. 31%), a significantly shorter LOH (53 h vs. 96 h), and less need of morphine-derived analgesics. No difference was found in R0 resections, and patients ultimately had higher quality-of-life assessments in the MILS group. In 2021, the OSLO-COMET trial published its long-term outcomes, which showed comparable results in terms of overall survival and RFS for both groups [92]. However, because of design flaws, small but possibly significant differences in long term outcomes must be taken into consideration and more trials may be necessary. In 2019, the LapOpHuva study [93] analyzed both short- and long-term outcomes for MILS or open liver resection (OLR). In the study, 204 patients were allocated to either MILS (101 patients) or OLR (103 patients). Significantly less morbidity and >IIIa D-C morbidity were seen in the OLR group (23.7% vs. 11.5%). Long-term OS and RFS were similar for both groups, with no significant differences.

The latest meta-analysis comprising 35 studies including the OSLO-COMET and the LapOpHuva studies compared MILS to open surgery for both simultaneous and staged resection of CRLM. Regarding staged resection, 23 studies were included in the analysis, showing comparable long-term results in OS and DFS, with no difference in R0 resections, and significantly shorter LOH and complication rates in favor of MILS [94]. When comparing simultaneous open surgery vs simultaneous MILS, 12 quality retrospective studies were assessed, with no available RCT. LOS was significantly decreased in simultaneous MILS, with no difference in R0 rates, OS, and DFS at 1, 3, or 5 years.

It is important to keep in consideration that as with any complex surgical technique, a learning curve and a high-volume load is needed to achieve satisfactory results. Chua et al. demonstrated that at least 25 and 50 surgeries for robotic and laparoscopic MILS respectively are needed to surpass the surgical learning curve [95]. Indeed, although MILS has been deemed feasible for a long time, there are recently published quality benchmarks based on over 11,000 patients every center should attempt to attain in order to offer the patients the best oncological results and the fewest possible postoperative complications [96].

The use of MILS for CRLM is another way that surgeons have been able to push the limits of what is feasible. MILS allows faster recovery and fewer complications, and therefore an improved quality of life and a larger number of patients who may benefit from a minimally invasive procedure. This has served as a less invasive option in patients with comorbidities.

## 7. Vascular Resections, Ex-Vivo and In-Situ Resections

As seen, limits in hepatic surgery are constantly being challenged. Advances in other fields such as liver transplant, particularly living donor liver transplant have equipped hepatic surgeons with an arsenal of techniques that may be applied in settings other than transplant.

Involvement of the main vessels supplying the inflow and outflow of the liver as well as the inferior vena cava (IVC) are not uncommon in CRLM. Vascular reconstruction and resection may limit CRLM surgery because of the prolonged hepatic ischaemia the patient may be subject to. Healthy hepatic tissue is known to be able to withstand approximately 60 min of warm ischemia without major repercussions [97,98,99,100,101], but damaged tissue such as cirrhotic, steatosis, and chemotherapy-induced steatohepatitis have lower tolerance to ischemia [100]. In addition, long-term results for these patients are usually poor because of the aggressiveness of the disease and the high chances of local recurrence.

Depending on the structure involved, total vascular isolation (TVI) may be required, greatly increasing the chances of irreversible ischemic injury. This is especially true for tumors in the hepatocaval confluence. However, TVI is seldom used in CRLM surgery, and the need to simultaneously divert flow from the portal vein, hepatic artery, and outflow structures is quite rare. In the exceptional case of large, central tumors with involvement of all hepatic veins, in which traditional in situ techniques cannot solve the issue, there may be room for ex vivo or in situ perfusion techniques to aid the complex vascular resections and reconstructions needed. These approaches may seem extremely aggressive, especially when compared with PSS, but considering there is no other alternative for such locally advanced tumors, surgery might be justified.

Involvement of the IVC can be divided into three groups: Segment I (infrarenal IVC), Segment II (suprarenal up to but not including the hepatic veins), and Segment III (from the hepatic veins upwards, with the possibility of cardiac extension). CRLM most frequently compromises the IVC at the hepatocaval confluence. In these cases, when R1vasc is not feasible, vascular resection reconstruction of the IVC and the confluence may be needed. In most cases, an anterior approach with parenchymal transection is used to allow for a better exposition of the area [102,103]. When the liver has been divided and the tumor is only attached to the IVC or the confluence, TVI is usually necessary. In a majority of cases, a veno-venous bypass is not required when performing a TVI. TVI allows resection of the affected vessels, and their reconstruction. Resection of 50% or more of the IVC usually mandates patching with either biological material (i.e., peritoneum or round ligament) or prosthetic replacement (18–20 mm ringed gore-tex graft) [104,105,106]. Hepatic veins can also be reconstructed either by venoplasty, graft interposition, or implantation in an IVC graft in order to ensure the adequate drainage of the future remnant [107,108,109,110]. Autologous venous grafts may even be harvested from the specimen itself.

When TVI is needed and predicted to last over 60 min, hypothermic perfusion techniques can aid in mitigating the damage caused by ischemia [99]. Initially described by Fortner et al. [111], hypothermic perfusion allows the surgeon to freely assess vascular involvement in a bloodless field and reconstruct the remaining structures without worrying about time limitation. Hypothermic perfusion may be performed in an in situ, ante situm and ex situ fashion. Briefly, in situ perfusion is perhaps the simplest of the techniques, in which mobilization of the liver followed by TVI and cold perfusion using preservation solutions through a venotomy in the PV and vented through the IVC. Venovenous bypass may be performed depending on the patient’s tolerance to TVI, but is not routinely required. The ante situm technique is almost identical to in situ perfusion, with a few differences. The ante situm technique requires extensive suprahepatic IVC dissection, including sometimes intrapericardial dissection, in order to double clamp, section, and sow back the suprahepatic IVC. This allows for greater access to the area of the hepatocaval confluence, with the ability to rotate the liver in a counterclockwise manner towards the anterior wall. The infrahepatic IVC can also be sectioned to allow for further hepatic mobilization. After resection, vascular reconstruction of the hepatic veins and IVC anastomosis are performed. Figure 7 shows a complex ante situm perfusion with IVC and hepatic vein reconstruction with both human and prosthetic grafts. Figure 8 shows partial caval resection and reconstruction with an autologous peritoneal graft.

Ex situ perfusion is the most complex technique of the three. Originally described by Pichlymayr et al. [112], the liver is explanted and perfused on the back table. The complete resection and complex venous reconstruction is performed before reimplantation of the liver, in a bloodless field. Indication for this procedure is very rare. Indeed, it is most likely suitable for patients deemed irresectable by other manners, in which complex reconstruction of the hepatic veins into the IVC and hilar resection and reconstruction are also planned. Its main setbacks include the need for additional biliary and vascular anastomoses, which may add morbidity to the procedure.

As to be expected, experience with ex situ perfusion is scarce compared with the evidence favoring in situ perfusion. This is mostly because of the fact that most resections are viable with less aggressive procedures, including in situ perfusion, and to the fact that many patients which are deemed irresectable by conventional methods are not properly referred for ex situ perfusion evaluation. The evidence in CRLM is even rarer. Azoulay et al. [113] reported on 77 patients who underwent ante situm perfusion for various tumors, of which 31 patients presented with CRLM. Thirty-day mortality for the entire cohort was 14%, far above the expected for routine hepatectomies. However, the 5-year OS for the CRLM group was 34%. Oldhafer et al. [114] reported six CRLM from a total of 22 patients who underwent ex vivo perfusion, with an OS of 21 months. However, that was prior to the major advances in chemotherapy previously described.

It is evident that these types of procedures are challenging and the patient selection criteria must be very strict in order to identify which patients will most likely benefit from such surgeries.

## 8. Transplantation for CRLM: Can We Justify the Effort?

What is perhaps the final boundary to push in CRLM surgery may seem a contradiction: the resection of non-resectable CRLM. Of course, the only way to achieve such a feat is by replacing the diseased liver with a new, disease-free organ. This was theorized many years ago. However, transplantation for CRLM has had several bumps on the road so far. The first one is a problem that traverses all patients in need of a liver transplant (LT): donor shortage. Organ shortage worldwide generates the need for allocation policies which consider “sickest-first” approaches. However, patients with malignant disease rarely present with liver failure and would not qualify using MELD scores, which is why in many countries patients are given ”exception“ points to level the field and elevate their waitlist priority [115]. On the other hand, in the earliest cases of LT for CRLM, discouraging results due to poor comprehension of the disease, lack of effective chemotherapy, and rapid recurrence became a major setback. The last “issue” associated with LT is the constant expansion of limits in resective surgery. What once was deemed inoperable is nowadays feasible by previously mentioned techniques. This has modified the criteria for candidates for LT in CRLM, and worse outcomes may result from bias in patient selection. An early series of patients subject to LT for CRLM had dismal results, with 5-year survival under 20% [116,117]. However, recent RCT (SECA-I [118] and SECA-II [119]) showed promising results for a select group of patients, with 5-year OS of 60% and 83%, respectively. These studies were able to determine four main risk factors that can detect poor candidates for LT in CRLM, which comprise the Oslo score diameter >5.5 cm, time from control of primary tumor <2 years, pre-LT CEA levels >80 ng/L, and disease progression while on chemotherapy. Current guidelines for patient selection suggest LT for CRLM in patients with unresectable disease, a correctly controlled primary tumor, favorable biology, and 6 months of sustained radiological response in patients in which the diagnosis of unresectable CRLM has an interval of at least 1 year [120]. A recent meta-analysis showed that from a total number of 110 patients analyzed in 18 studies, 5-year OS was 50.5% and 25% recurrence-free survival at 5 years. When considering cohorts after 2005 (stricter inclusion criteria), OS rose to 65% at 5 years. This is extremely encouraging, considering that the standard of care for patients with unresectable liver disease is systemic chemotherapy, with an OS at 5 years of under 10% with modern chemotherapy regimens [121]. There are several ongoing trials which attempt to determine which subgroup of patients will obtain the most benefit from a LT, thus providing an ethically responsible allocation. LT for CRLM is perhaps one of the biggest breakthroughs in surgical resection of colorectal metastasis. Transplant oncology is a field in constant expansion, which combines knowledge of surgical oncology and transplant surgery, such as the novel RAPID procedure [122], which combines the split donation procedure with TSH by removing the left lobe of the native liver, implanting a lateral left segment, and leaving the native right liver in situ until hypertrophy is sufficient.

## 9. Conclusions

There have been many advances regarding colorectal cancer and its biology. Patients who were deemed terminal 20 years ago are now considered resectable and have very satisfactory long-term survival. This is mainly because of the introduction of modern chemotherapy and biological agents, in combination with the cumulative surgical experience. Knowledge from other areas of liver surgery such as transplant surgery have assisted in making CRLM surgery a possibility for many patients, offering them excellent quality of life and survival. What once were considered limits, such as a small FLR, a synchronous disease, or great tumors which compromise the hepatocaval confluence or hilar structures, are now commonly approached in high-volume centers with optimal results. It is clear that limits in surgical resection of CRLM have been pushed and will most likely continue to be pushed with the introduction of new technologies and procedures, hopefully for the benefit of many more patients.

## Figures and Tables

**Figure 1 cancers-15-02113-f001:**
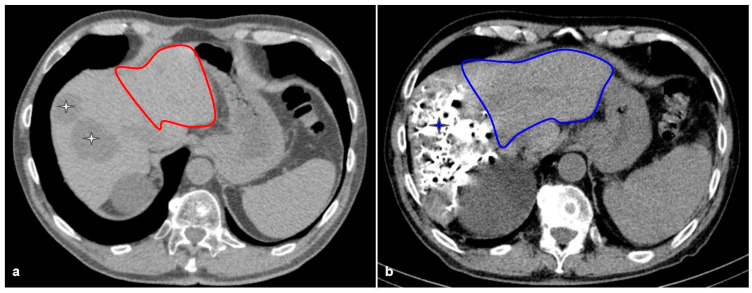
The case of a 70-year-old patient with synchronous bilobar CRLM. (**a**) CT volumetry of the FLR (red outline) is insufficient (FLR 20%, SPECT 20% and HIBA index of 8%); thus, a portal embolization with venous deprivation is performed. (**b**) The blue star marks the embolization material. The blue outline shows the exponential growth of the FLR after portal embolization (SPECT 58%, HIBA index 16%).

**Figure 2 cancers-15-02113-f002:**
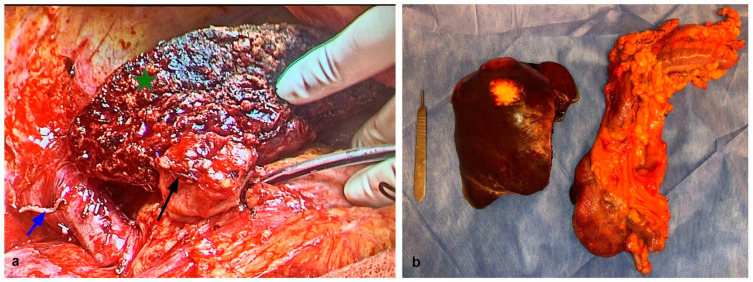
The patient of Figure 1 was subjected to a simultaneous approach after successful embolization. (**a**) The cut-edge surface of the hypertrophied FLR is marked with the green star, the IVC with the stapled right hepatic vein (blue arrow), and the sewn-over stump of the right portal branch with a black arrow. (**b**) The surgical specimens consist of right hepatectomy and a right colectomy, with a scalpel handle for scale.

**Figure 3 cancers-15-02113-f003:**
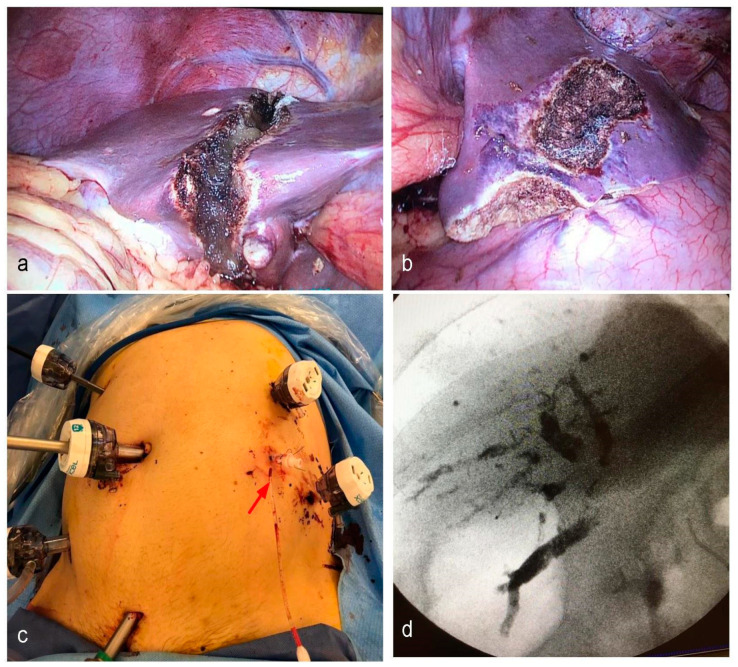
First stage of a simultaneous approach + Mini ALPPS in a patient with synchronous CRLM. (**a**) The transection line is shown on the right hemiliver. (**b**) The FLR has been cleared of multiple metastases. (**c**) The patient’s abdomen is shown for the sake of trocar placement. mini-ALPPS provides the benefit of a laparoscopic procedure, and can even assist in cannulation of the inferior mesenteric vein for PVE (red arrow). (**d**) Post-PVE portography that shows embolization material in all right portal branches.

**Figure 4 cancers-15-02113-f004:**
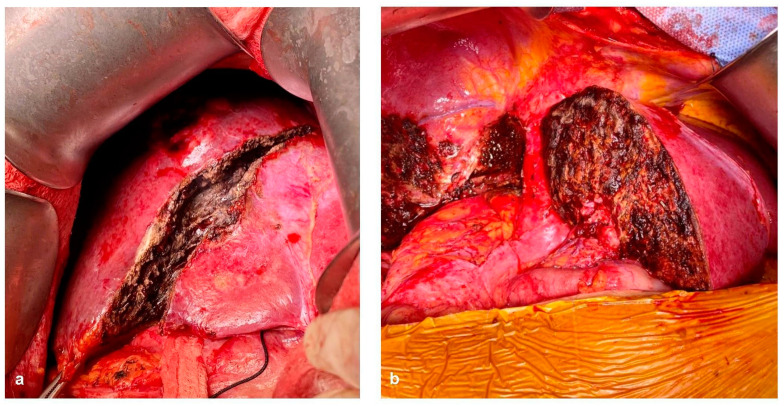
Second stage of a Mini-ALPPS in the patient previously described in Figure 3. (**a**) Ten days after a successful first stage, a conventional second stage was performed. Transection line and the deportalized right lobe can be observed. (**b**) Completion of the second stage with the cut edge of the FLR with neithr signs of bleeding nor bile leak.

**Figure 5 cancers-15-02113-f005:**
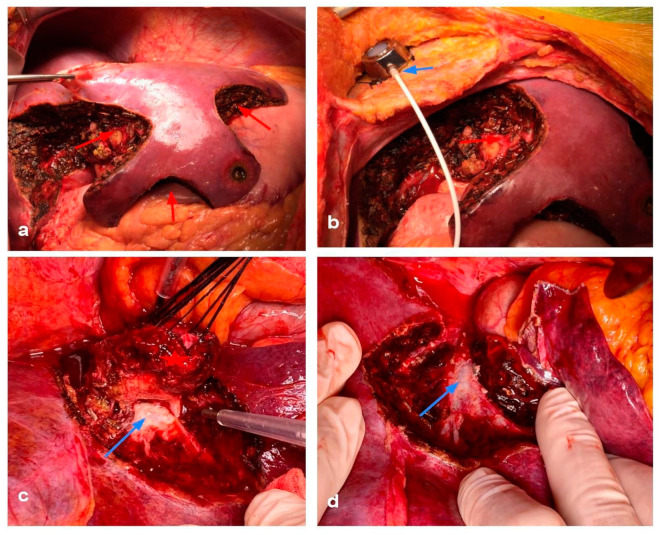
Parenchyma sparing surgery on a patient with multiple tumors. (**a**,**b**) The multiple bilobar resections with red arrows are shown. An intra arterial catheter for local chemotherapy was placed as well (blue arrow). (**c**) A tumor (red star) was in close proximity to the right hepatic vein (blue arrow). (**d**) It was removed with an R1 vasc approach, with preservation of the middle hepatic vein (blue arrow).

**Figure 6 cancers-15-02113-f006:**
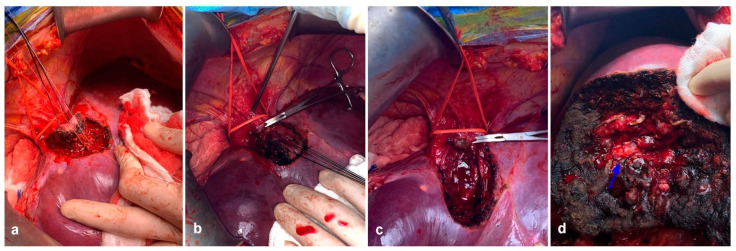
Parenchyma sparing surgery in a patient with a tumor located in the hepatocaval confluence. (**a**) Careful separation of the tumor from the surrounding tissue was performed. A red vascular loop was placed around the hepatic vein confluence in case clamping was required, and the tumor was marked by several stay sutures to allow for better exposure. (**b**) Due to the tumor’s proximity to the middle hepatic vein, a partial resection was necessary, and a Kelly clamp was used for hemostasis. (**c**) The tumor was completely removed, the middle hepatic vein has been ligated, and the right hepatic vein repaired using a patch of falciform ligament, as shown in (**d**) with a blue arrow.

**Figure 7 cancers-15-02113-f007:**
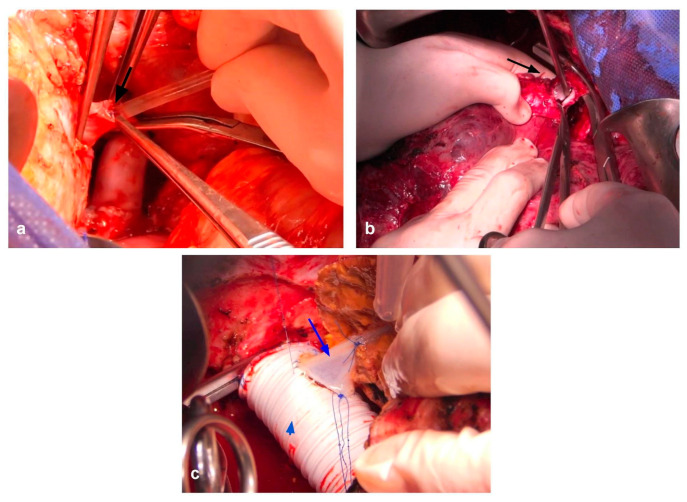
The ante situm procedure in a patient with a large CRLM invading the hepatocaval confluence. (**a**) Portotomy is performed and cannulated for cold perfusion with Wisconsin University Solution (black arrow). (**b**) The liver is rotated in a counterclockwise fashion to expose the hepatocaval confluence. A piggy-back type clamping of the hepatic veins is performed, and the left hepatic vein is opened (black arrow). (**c**) Reconstruction of the IVC with a prosthetic graft (arrowhead) and reimplantation of the left hepatic vein using a deceased donor iliac vein graft (blue arrow) are performed.

**Figure 8 cancers-15-02113-f008:**
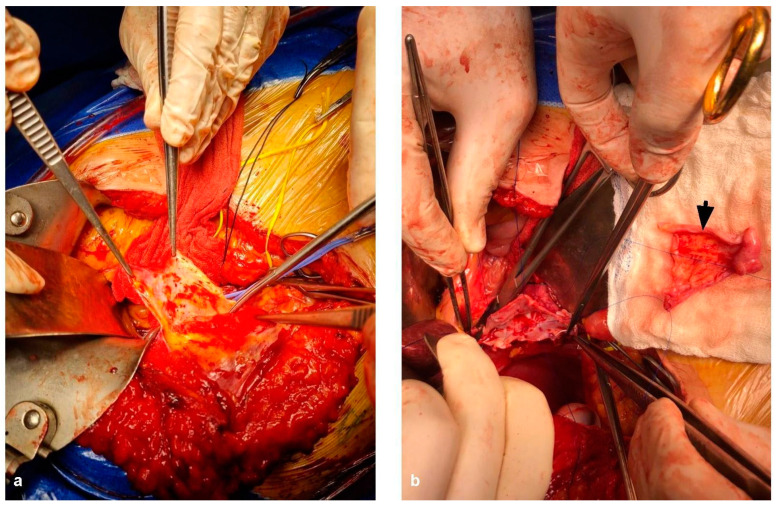
A patient with a large tumor invading the IVC. (**a**) The already resected and exposed endothelium of the IVC is shown. (**b**) The autologous peritoneal patch (black arrowhead), which will be used to close the defect, is shown. It should be noted that the epithelized side of the patch should always face inwards into the vascular lumen.

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
