# Peer review of "Pushing the Limits of Surgical Resection in Colorectal Liver Metastasis: How Far Can We Go?"

_cancers, 2023, doi:10.3390/cancers15072113_

Round 1

Reviewer 1 Report (Previous Reviewer 1)

The reviewers made several changes to the paper as suggested. I still share the drawback of reviewer 2 as previously stated - for surgeons the paper does not convey many noveltys. However, for the broad readership of "cancers" I consider the paper a well written summary of an important topic, additionally strenghtened by informative pictures and therefore would like to congratulate the authors for this nice work and suggest acceptance. 

Author Response

The authors would like to thank once again the reviewer 1 for their time and dedication in reviewing this manuscript, and apreciate the feedback. 

Reviewer 2 Report (New Reviewer)

Dear Authors, 

I am enthusiastic to review your manuscript, focused on a topic of interest to me, and based on current research in the field. I would like to take this chance to also congratulate you for your work and the effort put into the review of surgical resection limits in colorectal cancer liver metastasis

In the review process, I would like to point out aspects that might bring a slight improvement to your manuscript in the following section-by-section assessment:

The review would benefit from the addition of a Graphical Abstract scheme comprising of the most relevant data assessed by the authors.

The Abstract serves its purpose. 

Introduction:

Please reconsider the usage of the word “dramatic”[improvement] without citing values from original research studies. 

Sections 2-8:

Please provide the source of the images used. Shall they be taken out of another article, please cite the source and obtain permission from authors for usage. Shall they be original pictures taken by your research team, please provide an Ethical Committee approval for taking the pictures/using the CT scans. If neither of these could be provided, please replace all the pictures with schemes of the relevant aspects!

I consider the work well anchored in the current areas of interest and would like to support it for publication. 

Best regards, 

Your Reviewer

Author Response

Dear reviewer, 
The authors thank you for the time and dedication spent into reviewing this manuscript and producing thoughtful feedback for improvement of the manuscript. 

We have taken into consideration the reviewer's suggestion regarding a Graphical abstract, and have created one that briefly reflects the main topics of the paper and highlights its key points. We hope the reviewer finds it adequate.

Regarding the introduction, the authors agree with the reviewer and have removed the word " dramatic". In spite of available references, without citing concrete numbers, the word dramatic may be an overstep. 

Lastly, concerning authorship for the figures and images. All of them are property of the authors and have not been published previously or taken from any other manuscript. Upon admission in the hospital, informed consent is obtained from all patients in order to access their electronic medical records for unspecified research purposes in the future. 

We hope the reviewer finds these corrections sufficent, and are available for any further corrections deemed necesary. 
Thank you very much 

This manuscript is a resubmission of an earlier submission. The following is a list of the peer review reports and author responses from that submission.

Round 1

Reviewer 1 Report

The authors present a well written review of the surgical advances in colorectal liver metastases. 

The manuscript is written in a favourable style, however some minor flaws in orthography and shifts in tense could be corrected. 

In paragraph 3 (3. Two-staged hepatectomy (TSH), Associating liver partition with portal vein ligation for staged hepatectomy (ALPPS), and Mini-ALPPS: A way around extensive bilobar CRLM) the different ALPPS modifications could be described a bit better to clarify the differences. 

After paragraph 3, there is another paragraph 3 (3. How much margin do we actually need? vascular R1 (R1vasc), PSS and IOUS) - here I would suggest adding a few more references and strengthen the discussion and of course adjust the numbering. 

All in all, this is an important field and the strength of this review is its style, that is appealing to read and the nice images, that help visualizing the procedures. A small drawback that surgeons might not find much new information in the article, as all of most procedures can be considered established by now. 

Author Response

The authors would like to thank the reviewer for the time and dedication invested in reading and commenting the manuscript. 
We have run a new grammar check to correct the tense shifts throughout the paper. 
Regarding section 3: "Two-staged hepatectomy (TSH), Associating liver partition with portal vein ligation for staged hepatectomy (ALPPS), and Mini-ALPPS: A way around extensive bilobar CRLM", we have taken into consideration the recomendation and added a short paragraph detailing the differences between the ALPPS variations, for claryfication. It reads as follows:

" Briefly,  tourniquet ALPPS[47] included a vicryl tourniquet over the future transection line, in order to completely ligate communicating vessels between the FLR and the soon-to-be-removed right lobe. Microwave ablation ALPPS[48], consists of a laparoscopic ligation of the right portal vein, followed by microwave ablation (MWA) utilizing several antenna probes in the future transection line, in order to generate an “ avascular groove” that would separate the FLR from the embolized liver. Radiofrequency assisted ALPPS shares the same concept with MWA-ALPPS, by applying radiofrequency in the future transection line in order to achieve tissue necrosis and stimulation of growth without physical damage, thus limiting the number of complications in stage one. Partial-split ALPPS[49] is very similar to conventional ALPPS, with the difference that liver transection is not complete, and the in situ deportalized liver is only transected in a 50-80%.This was found to great reduce the morbidity of stage I, and allow a higher stage II completion."

Regarding the section titled " How much margin do we actually need? vascular R1 (R1vasc), PSS and IOUS ", we thank the reviewer for the apreciation. We have corrected the mistaken section numeration, and we have added several references and studies which compare PSS to conventional hepatic surgery for CRLM. the paragraph reads as such : 

"
There followed numerous reports favoring PSS over anatomical major hepatectomies from across the globe. These were coherent with initial reports, showing that PSS was better in terms of patient safety, but also found no difference in oncological short and long term results[64,65,66,67}. In 2016 Mise et. al[68] compared 300 patients who underwent either a PSS or a major hepatectomy for small (<30 mm) CRLM.  There was no difference in RFS between groups, but there was significantly higher 5 year survival in patients who underwent PSS. There were also significantly more patients who received a repeat hepatectomy upon liver-only recurrence in the PSS  vs. the major hepatectomy group, showing that not only survival favors PSS, but also that PSS does not preclude a repeat hepatectomy, expanding the amount of patients that can benefit from repeat procedures. A recent metaanalysis compared over 7000 who received either a PSS or major hepatectomy throughout 18 different studies[68}. PSS was superior to major hepatectomies in terms of perioperative morbidity, and there was no difference in terms of OS, RFS and surgical margins. This means that PSS can safely acquire R0 margins, without oncological compromise of the patient, and avoidance of major hepatectomies.

PSS, IOUS and R1 Vasc push the limits in terms of what we perceive as unresectable disease, and may benefit many patients. However, there are a few issues: The technical difficulties and expertise necessary for the correct use of IOUS, the extended surgical times, the apparent “setback” regarding the approach (PSS most of the time requires open surgery) and the lack of randomized controlled trials comparing PSS to major hepatectomies remain an issue." 

We believe the reviewer will find these corrections adequate, and are available for any additional corrections needed. This narrative review is intended to convey the findings which have "pushed the limits" of hepatic surgery  in CRLM to the general medical audience, which may or may not posses limited knowledge of these techniques. It is true that most of these techniques are well known to established HPB surgeons, but perhaps not so much to general surgeons or surgeons who are yet in training. We thank the reviewer once again.

Reviewer 2 Report

Dear Author, thank you for you submission to cancers concerning a such interesting and actual item.

Despite the topic you treat is very important for the scientific community and your center is a referral center for liver surgery, there are some drawbacks in the paper.

What you report in your review is very well known either about techniques either about prognosis so I do suggest to improve your articles considering the real hot topics in CRLM, like indications to simoultaneous surgery, the needs of one or two equipes to best manage colorectal and liver surgery. Th real impact of minimally invasive approaches concerning indications to simoultaneous surgery. Furthermore tehre technical aspetcs that you did not consider like the optimal trocar placement or the standard of care about hospital stay, complications, and oncological outcomes.

On the other hand I suggest performing a sistyematic review about CRLM surgical management.

Author Response

The authors would like to thank the reviewer for his time and thorough examination of the manuscript. We are aware that some of the topics adressed in this manuscript are the standard of care in many institutions worldwide. However,  this manuscript was submittted for a special request from the editors as a narrative review, which is meant to summarize and analyze the existing data regarding the advances that have allowed surgical resection of CRLM to greatly improve and reach many more patients than years ago. As a high impact factor journal, "Cancers" is read by many different specialties and physicians in formation, thus the plan was to reach as many as posible. 
However, we have indeed considered your suggestion, and have added a section regarding the use of Minimally invasive liver surgery as another important milestone in pushing the limits of CRLM surgery, right after the simultaneous surgery section, which discusses the indications, safety and comparison to open surgery. Although al the topics suggested are important and "hot topic", we believe that they might exceed the reaches of a narrative review such as this one, as well as the metaanalysis for the management of CRLM, which is extremely complex. 
We hope the reviewer finds these corrections sufficent, and are open for any more suggestions. 
Thank you  very much 

Round 2

Reviewer 2 Report

Dear Authors, thank you for your kind reply, however as you written in your letter you modified the paper only concerning the minimally invasive approach, and more in detail you treated only few studies about the laparoscopic and robotic approach.

So, in my opinion,  the paper is not so relevant for the scientific community to be acceptable